# LINEAR VIDEO TRANSFORMER WITH FEATURE FIXATION

## ABSTRACT

Vision Transformers have achieved impressive performance in video classification, while suffering from the quadratic complexity caused by the Softmax attention mechanism. Some studies alleviate the computational costs by reducing the number of tokens in attention calculation, but the complexity is still quadratic. Another promising way is to replace Softmax attention with linear attention, which owns linear complexity but presents a clear performance drop. We find that such a drop in linear attention results from the lack of attention concentration on critical features. Therefore, we propose a feature fixation module to reweight feature importance of the query and key before computing linear attention. Specifically, we regard the query, key, and value as various latent representations of the input token, and learn the feature fixation ratio by aggregating Query-Key-Value information. This is beneficial for measuring the feature importance comprehensively. Furthermore, we enhance the feature fixation by neighborhood association, which leverages additional guidance from spatial and temporal neighbouring tokens. The proposed method significantly improves the linear attention baseline and achieves state-of-the-art performance among linear video Transformers on three popular video classification benchmarks. With fewer parameters and higher efficiency, our performance is even comparable to some Softmax-based quadratic Transformers.

## 1 INTRODUCTION

Vision Transformers [19, 86, 49, 82, 15] have been successfully applied in video processing. Recent video Transformers [57, 4, 7, 6, 20] have achieved remarkable performance on challenging video classification benchmarks, *e.g.*, Something-Something V2 [25] and Kinectics-400 [10]. However, they always suffer from quadratic computational complexity, which is caused by the Softmax operation in the attention module [77, 93]. The quadratic complexity severely constrains the application of video Transformers, since the task of video processing always requires to handle a huge amount of input tokens, considering both the spatial and temporal dimensions.

Most of the existing efficient designs in video Transformers attempt to reduce the number of tokens attended in attention calculation. For example, [4, 6] factorize spatial and temporal tokens with different encoders or multi-head self-attention modules, to only deal with a subset of input tokens. [7] calculates the attention only from the target frame, *i.e.*, space-only. [57] restricts tokens to a spatial neighbourhood that can reflect dynamic motions implicitly. Despite reducing the computational costs, they still have the inherent quadratic complexity, which prohibits them from scaling up to longer input, *e.g.*, more video frames or larger resolution [6, 57].

Another practical way, widely used in the Natural Language Processing (NLP) community, is to decompose Softmax with certain kernel functions and linearize the attention via the associate property of matrix multiplication [60, 35, 14]. Recent linear video Transformers [57, 93] achieve higher efficiency, but present a clear performance drop compared to Softmax attention. We find that the degraded performance is mainly attributed to the lack of attention concentration to critical features in linear attention. Such an observation is consistent with the concurrent works in NLP [60, 85].

Accordingly, we propose to concentrate linear attention on critical features through **feature fixation**. To this end, we reweight the feature importance of the query and key prior to computing linear attention. Inspired by the idea of classic Gaussian pyramids [53, 43] and modern contrastive learning [64, 72], we regard the query, key, and value as latent representations of the input token. Latent

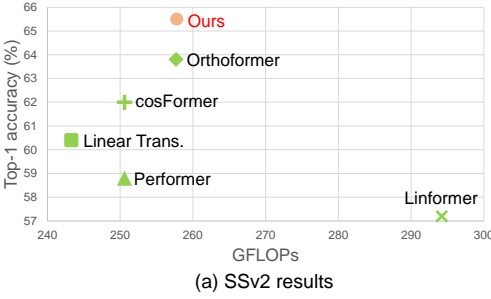 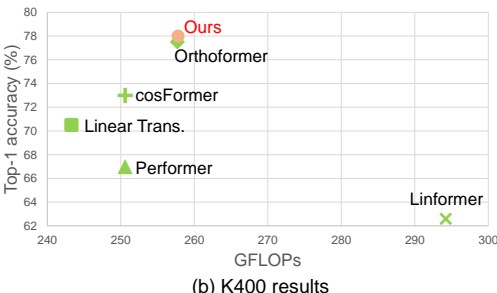

(a) SSv2 results            (b) K400 results

Figure 1: Video classification performance of linear Transformers. We report Top-1 accuracy and GFLOPs of the state-of-the-art linear video Transformers on (a) SSv2 and (b) K400 datasets. Our model achieves the best accuracy among its counterparts.

space projection will not destroy the image structure, *i.e.*, salient features are expected to be discriminative across all spaces [71]. Therefore, we aggregate Query-Key-Value features when generating the fixation ratio, which is beneficial for measuring the feature importance comprehensively.

Meanwhile, the salient feature activations are usually locally accumulative [11, 67]. In the continuous video, critical information contained in the target token could be shared by its spatial and/or temporal neighbour tokens. We hence propose to use **neighbourhood association** as extra guidance for feature fixation. Specifically, we reconstruct each key and value vectors (they are responsible for information exchange between tokens [29, 7]) by sequentially mixing key/value features of nearby tokens in the spatial and temporal domain. This is efficiently realized by employing the feature shift technique [84, 91]. Experimental results demonstrate the effectiveness of feature fixation and neighbourhood association in improving the classification accuracy.

Our primary contributions are summarized as follows: **1)** We discover that the performance drop of linear attention results from not concentrating attention distribution on discriminative features in linear video Transformers. **2)** We develop a novel linear video Transformer with feature fixation to concentrate salient attention. It is further enhanced by neighbourhood association, which leverages the information from nearby tokens to measure the feature importance better. Our method is simple and effective, and can also be seamlessly applied to other linear attention variants and improve their performance. **3)** Our model achieves state-of-the-art results among linear video Transformers on popular video classification benchmarks (see Fig. 1), and its performance is even comparable to some Softmax-based quadratic Transformers with fewer parameters and higher efficiency.

## 2 RELATED WORK

**Video classification by CNNs.** CNN-based video networks are normally built upon 2D image classification models. To aggregate temporal information, current works either fuse the features of each video frame [80, 45], or build spatial-temporal relations through 3D convolutions [28, 62, 73] or RNNs [18, 40, 42]. Although the latter ones have achieved state-of-the-art performance, they suffer from being significantly more computationally and memory-intensive. Several techniques have been proposed to enhance efficiency, *e.g.*, temporal shift [45, 51], adaptive frame sampling [37, 3, 88, 87], and spatial/temporal factorization [74, 32, 75, 21].

**Video Transformers.** Vision Transformers [19, 86, 49, 82, 15] have achieved a great success in computer vision tasks. It is straightforward to extend the idea of vision Transformers to video processing, considering both the spatial and temporal dependencies. However, the full spatial-temporal attention is computationally expensive, and hence several video Transformers try to reduce the costs via factorization. TimeSformer [6] exploits five types of factorization, and empirically finds that the divided attention, where spatial and temporal features are separately processed, leads to the best accuracy. A similar conclusion is drawn in ViViT [4], which separates spatial and temporal features with different encoders or multi-head self attention (MHSA) modules. [7] introduces a local temporal window where features at the same spatial position are mixed. Some other methods have also explored dimension reduction [24, 34], token selection [78, 2], local-global attention stratification [44], deformable attention [77], temporal down-sampling [65, 92], locality strengthening [50, 23], hierarchical learning [90, 89], multi-scale fusion [20, 26, 83], and so on.

**Efficient Transformers.** The concept of efficient Transformers was originally introduced in NLP, aiming to reduce the quadratic time and space complexity caused by the Transformer attention [39, 69, 35, 36, 81]. The mainstream methods use either patterns or kernels [70]. Pattern-based approaches [61, 46, 5, 12] sparsify the Softmax attention matrix with predefined patterns, *e.g.*, computing attention at fixed intervals [5] or along every axis of the input [30]. Though benefiting from the sparsity, they still suffer from the quadratic complexity to the input length. By contrast, kernel methods [35, 59, 14, 13, 60] reduce the complexity to linear by reformulating the attention with proper functions. As suggested by [60], these functions should be decomposable and non-negative.

## 3 PROBLEM FORMULATION

In this section, we first formulate the video Transformer and analyze their quadratic complexity in Section 3.1. Linear attention is introduced in Section 3.2 and its fundamental shortcoming in the performance drop is discussed in Section 3.3.

### 3.1 VIDEO TRANSFORMER

A video clip can be denoted as $\mathcal{V} \in \mathbb{R}^{T \times H \times W \times C}$, containing $T$ frames with a resolution of $H \times W$ and a channel number of $C$. The video Transformer maps each $H \times W$ frame to a sequence of $S$ tokens, and hence has $N = T \times S$ tokens in total. The embedding dimension of a token is $D$. These tokens are further projected into the query $\mathbf{Q} \in \mathbb{R}^{N \times D}$, key $\mathbf{K} \in \mathbb{R}^{N \times D}$, and value $\mathbf{V} \in \mathbb{R}^{N \times D}$. The self-attention is defined as[1]

$$\mathbf{Y} = \delta(\mathbf{Q}\mathbf{K}^{\mathrm{T}})\mathbf{V}, \tag{1}$$

where $\delta(\cdot)$ is a similarity function, *e.g.*, Softmax in standard Transformers [76, 19]. As shown, $\mathbf{Q}\mathbf{K}^{\mathrm{T}}$ has a shape of $N \times N$. Therefore, the use of Softmax will lead to quadratic complexity with respect to the input length, *i.e.*, $\mathcal{O}(N^2)$. This is computationally expensive especially for a long sequence of high-resolution videos. Existing efficient designs in video Transformers [4, 7, 6, 57] mostly focus on reducing the number of attended tokens when calculating the attention, which, however, still suffer from the quadratic complexity. Another promising way is to approximate Softmax in a linear form [57, 93], but it is less comparable in accuracy than quadratic Transformers and always time-consuming in practical use due to the sampling operation.

### 3.2 LINEAR ATTENTION

To reduce the complexity close to linear $\mathcal{O}(N)$, linear Transformers [35, 59, 14, 13, 60, 81, 66] decompose the similarity function $\delta(\cdot)$ to a kernel function $\rho(\cdot)$, where $\delta(\mathbf{Q}\mathbf{K}^{\mathrm{T}}) = \rho(\mathbf{Q})\rho(\mathbf{K})^{\mathrm{T}}$. The associative property of matrix multiplication allows to multiply $\rho(\mathbf{K})^{\mathrm{T}}$ and $\mathbf{V}$ first without affecting the mathematical result, *i.e.*,

$$\mathbf{Y} = (\rho(\mathbf{Q})\rho(\mathbf{K})^{\mathrm{T}})\mathbf{V} = \rho(\mathbf{Q})(\rho(\mathbf{K})^{\mathrm{T}}\mathbf{V}), \tag{2}$$

where the shape of $(\rho(\mathbf{K})^{\mathrm{T}}\mathbf{V})$ is $D \times D$. By changing the order of matrix multiplication, the complexity is reduced from $\mathcal{O}(N^2D)$ to $\mathcal{O}(ND^2)$. Since we always have $N \gg D$ in practice, $\mathcal{O}(ND^2)$ is approximately equal to $\mathcal{O}(N)$, *i.e.*, growing linearly with the sequence length. As discussed in [60, 8], a simple ReLU [1] is a good candidate for the kernel function, which satisfies the requirement of being non-negative and easily decomposable. The attended output is formulated with row-wise normalization, *i.e.*,

$$\mathbf{Y}_i = \frac{\mathrm{ReLU}(\mathbf{Q}_i)\sum_{j=1}^{N}\mathrm{ReLU}(\mathbf{K}_j)^{\mathrm{T}}\mathbf{V}_j}{\mathrm{ReLU}(\mathbf{Q}_i)\sum_{j=1}^{N}\mathrm{ReLU}(\mathbf{K}_j)^{\mathrm{T}}}. \tag{3}$$

### 3.3 ACHILLES HEEL OF LINEAR ATTENTION

Despite the lower computational complexity, linear attention always presents a clear performance drop compared to Softmax [8, 68, 60]. We also observe this drop when applying the linear attention to video Transformers, as shown in Table 1 and 2. The lack of non-linear attention concentration

---

[1]For simplicity, we focus on the single head and ignore the scaling term $\sqrt{D}$ that normalizes $\mathbf{Q}\mathbf{K}^{\mathrm{T}}$.

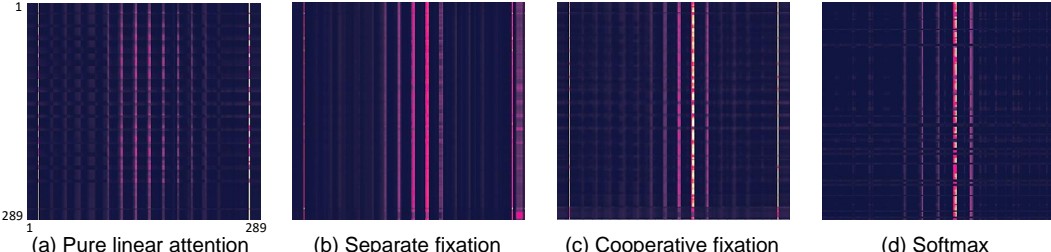

| (a) Pure linear attention | (b) Separate fixation | (c) Cooperative fixation | (d) Softmax |

Figure 2: Attention matrix visualization. The data are extracted from Layer 11 of a validation example in SSv2, and brighter color indicates larger values. (a) Pure linear attention cannot properly concentrate on high attention scores like (d) Softmax. (b) After employing separate fixation to query and key features, the attention becomes more concentrated on salient tokens. (c) With cooperative feature fixation, the attention is further concentrated and closer to that of Softmax.

in the process of dot-product normalization is considered to be the primary cause [60, 8]. From Fig. 2(a), we observe that the distribution of linear attention is smoother, *i.e.*, it cannot concentrate on high attention scores as that in Softmax (Fig. 2(d)). The non-concentration would incur the existence of noisy/irrelevant information, *e.g.*, a number of attention scores have been unexpectedly highlighted in Fig. 2(a) compared with Softmax. This, in turn, would aggravate the diffusion of linear attention. In this work, we propose to improve the concentration in linear attention. Specifically, we aim to inhibit the non-essential parts of $\mathbf{Q}$ and $\mathbf{K}$[2] and highlight the role of discriminative features to sharpen the results of linear attention, as discussed in Section 4.

## 4 METHOD

In this section, we will introduce the idea of feature fixation (Section 4.1 and Section 4.2) and neighborhood association (Section 4.3). Feature fixation reweights the feature importance of the query and key, and the cooperative fixation encourages different latent representations of a token to collectively determine the feature importance. Neighborhood association makes use of the features of nearby tokens as extra guidance for feature fixation.

### 4.1 SEPARATE FEATURE FIXATION

To highlight the essential parts of $\mathbf{Q}$ and $\mathbf{K}$, their features should be reweighted in a non-linear manner, *i.e.*, keeping important features almost unchanged (assigning weights closer to 1) while down-weighting (closer to 0) the non-essential activation. Inspired by the excitation strategy [31] and context gating mechanism [55] in CNNs, we recalibrate the query and key in the feature dimension (see Fig. 3(a) and Appendix A.2). For simplicity, we denote $\rho(\mathbf{Q})$ and $\rho(\mathbf{K})$[3] as $\check{\mathbf{Q}}$ and $\check{\mathbf{K}}$ respectively. The recalibrated results are

$$\gamma_q = \sigma(\varphi_q(\check{\mathbf{Q}})), \gamma_k = \sigma(\varphi_k(\check{\mathbf{K}})), \hat{\mathbf{Q}} = \gamma_q \odot \check{\mathbf{Q}}, \hat{\mathbf{K}} = \gamma_k \odot \check{\mathbf{K}}, \tag{4}$$

where $\varphi_q(\cdot), \varphi_k(\cdot)$: $\mathbb{R}^{N \times D} \to \mathbb{R}^{N \times D}$ learn a nonlinear interaction between feature channels with $D \times D$ weights, for measuring feature importance [31]. $\sigma(\cdot)$ is the element-wise Sigmoid function individually performed on each feature dimension, so that $\gamma_q, \gamma_k \in \mathbb{R}^{N \times D}$ are fixation ratios (0-1) of $\mathbf{Q}$ and $\mathbf{K}$ respectively. $\odot$ represents the element-wise multiplication. Fig. 2(b) shows that after reweighting $\mathbf{Q}$ and $\mathbf{K}$ separately, the attention becomes more concentrated. However, it still has an obvious gap with Softmax, *i.e.*, the concentration on the most essential tokens is not sufficiently sharpened. Though separately learning the fixation ratio is beneficial, it ignores the internal connection of $\mathbf{Q}$ and $\mathbf{K}$ that is critical to the comprehensive measurement of feature importance.

---

[2]This is because the attention matrix is calculated from $\mathbf{Q}$ and $\mathbf{K}$. Even though the $N \times N$ matrix does not exist in linear attention, $\mathbf{Q}$ and $\mathbf{K}$ still take effect in normalization in Eq. 3.

[3]We empirically find that directly reweighting the original $\mathbf{Q}$ and $\mathbf{K}$ without ReLU activation leads to OOM in network training.

Figure 3: Illustration of feature fixation. (a) Separate fixation learns the reweighting ratios of the query and key individually. (b) Cooperative fixation facilitates the communication among the query, key, and value, enabling a more comprehensive understanding of feature importance.

## 4.2 COOPERATIVE FEATURE FIXATION

The feature fixation ratios in Eq. 4 separately come from $\mathbf{Q}$ and $\mathbf{K}$, which has the risk of deficiently measuring the feature importance. In fact, the attention mechanism in Transformers naturally projects each input token to three representations, *i.e.*, $\mathbf{Q}$, $\mathbf{K}$, and $\mathbf{V}$. Inspired by modern contrastive learning [64, 72] and classic Gaussian Pyramid [53, 43], projecting an image to different latent spaces would not destroy the image structure. For instance, in contrastive learning, an image is projected to various spaces by random augmentation operations, while the latent features of these augmented images are expected to share similar properties, both locally and globally [71]. Besides, take SIFT [53] as an example of Gaussian Pyramid methods. Though an image is processed by different Gaussian kernels, its salient pixels (*i.e.*, keypoints in SIFT) have a high response across all the Gaussian spaces.

Therefore, we propose to obtain the feature fixation ratio by facilitating the cooperation among $\mathbf{Q}$, $\mathbf{K}$, and $\mathbf{V}$. This is motivated by the above insight that the essential features of a token should be discriminative among all its three latent spaces, and vice versa for unimportant information. To this end, we adjust Eq. 4 as follows:

$$\gamma = \sigma\left(\varphi(\check{\mathbf{Q}}, \check{\mathbf{K}}, \check{\mathbf{V}})\right), \hat{\mathbf{Q}} = \gamma \odot \check{\mathbf{Q}}, \hat{\mathbf{K}} = \gamma \odot \check{\mathbf{K}}, \tag{5}$$

where $\varphi(\cdot)$ supplies a cooperation space of $\mathbf{Q}$, $\mathbf{K}$, $\mathbf{V}$ for the purpose of comprehensively estimating the importance degree of each feature channel. Their cooperation, illustrated in Fig. 3(b), can be achieved by aggregating their features, *e.g.*, concatenation (used in our implementation) and addition. Moreover, we keep the fixation ratio identical when reweighting the query and key, which is in line with our design philosophy that feature importance should be consistent across various spaces. Fig. 2(d) shows that using our cooperative feature fixation leads to sharper attention concentration than separate learning of each fixation ratio. In Section 5 we prove that the enhanced concentration can yield better accuracy.

## 4.3 NEIGHBORHOOD ASSOCIATION

The vision features are usually continuous in temporal [11, 41] and spatial [67, 58] neighbourhood. Analogously, the prediction confidence increases if all the tokens within a local region can reach a consensus for feature importance. Therefore, it is straightforward to introduce the features of temporal and spatial neighbour tokens as additional guidance. To enhance the communication within the neighbourhood, we take advantage of the feature shift technique [84, 29, 7] that is parameter-free and efficient in use. Considering that each query is attended by key-value pairs from other tokens [29, 7] for information exchange, each $\mathbf{K}$ and $\mathbf{V}$ is reconstructed by sequentially mixing its temporally and spatially adjacent tokens including itself (namely, *temporal shift* and *spatial shift*). The neighbourhood association is proven to be effective in improving the classification accuracy. Technical details can be found in Appendix A.3.

## 5 EXPERIMENTS

In this section, we conduct extensive experiments to verify the effectiveness of our linear video Transformer. We start with experimental settings in Section 5.1, *e.g.*, the backbone, datasets, and training/inference details. Subsequently, we show the competitive performance of our model to SOTA methods in Section 5.2. The importance of each module and more comprehensive model analysis are discussed in Section 5.3.

Table 1: Comparison with the state-of-the-art on SSv2. $\times$T represents the number of input frames. Our model achieves the best accuracy (**bolded**) among linear video Transformers and its performance is also comparable to Softmax-based Transformers and CNN methods.

| | Method | Top-1 | Top-5 | $\times$T | # Par. (M) | Views | GFLOPs |
|---|---|---|---|---|---|---|---|
| **CNN** | SlowFast [22] | 63.1 | 87.6 | 8 | 53.3 | $1 \times 3$ | 106.0 |
| | TSM [45] | 63.3 | 88.2 | 16 | 42.9 | $2 \times 3$ | 62.0 |
| | MSNet [38] | 64.7 | 89.4 | 16 | 24.6 | $1 \times 1$ | 67.0 |
| **Softmax** | TSformer-HR [6] | 62.5 | - | 16 | 121.4 | $1 \times 3$ | 1703.0 |
| | MViT-B [20] | 64.7 | 89.2 | 16 | 33.6 | $1 \times 3$ | 70.5 |
| | ViViT-L [4] | 65.4 | 89.8 | 32 | 352.1 | $4 \times 3$ | 3992.0 |
| | XViT [7] | 66.2 | 90.6 | 16 | 92.0 | $1 \times 3$ | 850.0 |
| | Mformer [57] | 66.5 | 90.1 | 8 | 109.0 | $1 \times 3$ | 369.5 |
| **Linear** | Linformer [81] | 57.2 | 84.4 | 16 | 53.5 | $2 \times 3$ | 294.2 |
| | Performer [14] | 58.8 | 85.4 | 16 | 51.7 | $2 \times 3$ | 250.6 |
| | Linear Trans. [35] | 60.4 | 86.7 | 16 | 51.7 | $2 \times 3$ | 243.3 |
| | cosFormer [60] | 62.0 | 87.3 | 16 | 51.7 | $2 \times 3$ | 250.6 |
| | RALA [93] | 63.7 | - | 16 | 102.0 | $1 \times 3$ | 257.6 |
| | Oformer [57] | 63.8 | - | 16 | 102.0 | $1 \times 3$ | 257.7 |
| | **Ours** | **65.5** | **90.1** | 16 | 52.2 | $2 \times 3$ | 257.8 |

## 5.1 EXPERIMENTAL SETUP

**Backbone.** We use the standard ViT architecture [19] as the backbone. Our default model, built on top of XViT [7] and pretrained on ImageNet-21k [17], has the embedding dimension of 512, patch size of 16, 12 Transformer layers each with 8 attention heads. We employ the factorized spatial-temporal approach for calculating attention, which is shown to be effective in [6, 4, 49] owing to the distinct learning in spatial and temporal dimensions.

**Datasets.** We evaluate our method on three popular video action classification datasets. *Something-Something V2* (SSv2) [25], focusing more on temporal modelling, consists of $\sim$169k training and $\sim$24.7k validation videos in 174 classes. *Kinectics-400* (K400) [10] contains $\sim$240k videos for training and $\sim$20k for validation with 400 human action classes. *Kinectics-600* (K600) [9] is extended from K400 with $\sim$370k training and $\sim$28k validation videos containing 600 categories.

**Evaluation metrics.** When comparing accuracy, we report Top-1 and Top-5 accuracy (%) on SSv2, K400 and K600. We use GFLOPs for measuring computational costs as in most video classification works, and also provide the throughput (vps: videos per second) when available to reveal the actual processing speed.

**Training phase.** By default, we follow a similar training and augmentation strategy to XViT [7], *e.g.*, a $16 \times 224 \times 224$ video input, SGD with the momentum as 0.9, and autoaugment [16] for data augmentation. The base learning rate is 0.035 and scheduled with the cosine policy [52]. We train the models with 35 epochs (5 for warm-up) and a batch size of 32 on 8 V100 GPUs using PyTorch [56]. Detailed training configuration is listed in Appendix A.1.

**Inference phase.** Following previous works [7, 6, 4, 57], we divide test videos into $x \times y$ views, *i.e.*, $x$ is the number of uniformly-sampled clips in the temporal dimension and $y$ represents the number of spatial crops, and obtain the final prediction by averaging the scores over all views. The views of competing methods are listed in Table 1. See Appendix A.4 for the impact of views.

## 5.2 MAIN RESULTS

**Methods to compare.** We compare our model with six state-of-the-art linear attention variants, *i.e.*, Oformer [57], LARA [93], Linformer [81], Performer [14], Linear Trans. [35], and cosFormer [60]. For Oformer and LARA, we report the results from their papers. The remaining four are proposed in the NLP literature, so we re-implement them in our backbone and train with the same settings as the proposed linear Transformer. We also compare representative Softmax-based quadratic Transformers and CNN methods *for reference only*.

Table 2: Comparison with the state-of-the-art on K400 and K600. Our model achieves the best accuracy (**bolded**) among linear video Transformers and its performance is also comparable to Softmax-based Transformers and CNN methods.

| Method | | **K400** | | | **K600** | | |
|---|---|---|---|---|---|---|---|
| | | Top-1 | Top-5 | Views | Top-1 | Top-5 | Views |
| **CNN** | I3D [9] | 71.1 | 90.3 | - | 71.9 | 90.1 | - |
| | X3D-M [21] | 76.0 | 92.3 | $10 \times 3$ | 78.8 | 94.5 | $10 \times 3$ |
| | SlowFast [22] | 77.0 | 92.6 | $10 \times 3$ | 79.9 | 94.5 | $10 \times 3$ |
| **Softmax** | TSformer [6] | 78.0 | 93.7 | $1 \times 3$ | 79.1 | 94.4 | $1 \times 3$ |
| | Mformer [57] | 79.9 | 94.2 | $10 \times 3$ | 81.6 | 95.6 | $10 \times 3$ |
| | MViT-B [20] | 80.2 | 94.4 | $1 \times 5$ | 83.8 | 96.3 | $1 \times 5$ |
| | XViT [7] | 80.2 | 94.7 | $2 \times 3$ | 84.5 | 96.3 | $1 \times 3$ |
| | ViViT-L [4] | 80.6 | 94.7 | $4 \times 3$ | 83.0 | 95.7 | $4 \times 3$ |
| **Linear** | Linformer [81] | 62.6 | 86.0 | $4 \times 3$ | 67.5 | 88.5 | $4 \times 3$ |
| | Performer [14] | 67.0 | 88.8 | $4 \times 3$ | 72.0 | 91.1 | $4 \times 3$ |
| | Linear Trans. [35] | 70.5 | 90.2 | $4 \times 3$ | 74.5 | 92.3 | $4 \times 3$ |
| | cosFormer [60] | 73.0 | 91.1 | $4 \times 3$ | 78.4 | 92.5 | $4 \times 3$ |
| | RALA [93] | 77.5 | - | $10 \times 3$ | - | - | - |
| | Oformer [57] | 77.5 | - | $10 \times 3$ | - | - | - |
| | **Ours** | **78.0** | **94.2** | $4 \times 3$ | **84.1** | **96.2** | $4 \times 3$ |

**SSv2.** Quantitative results on SSv2 are reported in Table 1. We achieve the best Top-1 accuracy among linear models, and surpass the second best, *i.e.*, Oformer [57], by 1.7% with nearly halved parameters. Remarkably, we outperform several SOTA Softmax-based video Transformers, *e.g.*, TSformer [6], ViViT [4], with significantly smaller FLOPs (>6 times). Our performance is also comparable to XViT [7] and Mformer [57], *e.g.*, our Top-5 (90.1%) is identical to that of Mformer. CNN methods are generally less competitive although they tend to be faster with lighter networks.

**K400 & K600.** Our performance on K400 and K600 is consistent with that on SSv2, *i.e.*, outperforming linear models and CNN methods to a clear margin, and being comparable to Softmax-based Transformers. We achieve notably more competitive results on K600, *e.g.*, only having a 0.4% Top-1 gap compared with the best quadratic Transformer XViT [7].

## 5.3 ABLATION STUDY AND MODEL ANALYSIS

We ablate primary design choices and analyze our model on SSv2 as it is a more challenging dataset containing fine-grained motion cues [7, 57].

**Order of feature fixation and neighbourhood association.** From a broad view, we start with the order of the two primary modules in our model. We find that neighbourhood association first produces better accuracy than feature fixation first, *i.e.*, 0.82% and 0.42% gain in Top-1 and Top-5 accuracy respectively. This is reasonable because the shifted features from other tokens can be further refined by the feature fixation mechanism. In the following, our default setting is always *neighbourhood association prior to feature fixation* unless otherwise specified.

**Effect of neighbourhood association.** Table 3 reports the results of applying spatial and temporal feature shifts to the baseline model with pure linear attention in Eq. 3. The spatial shift brings a 0.32% gain in Top-1 accuracy while the temporal shift improves the performance by 2.51%. This is because the SSv2 dataset is generally more dependent on temporal information (we also find that the spatial shift is more helpful to K400 in Appendix A.5). The results indicate the effectiveness of neighbourhood association to aggregate more features from spatially and temporally nearby tokens.

**Effect of feature fixation.** In Table 3, we observe that feature fixation improves the baseline by 0.90%. Although the gain is smaller than the temporal shift[4], it has its distinctive usefulness, *i.e.*, reweighting feature importance of the token itself and the features from other tokens. The combination of feature fixation and neighbourhood association yields the best accuracy (surpassing the baseline by 3.87%).

---

[4]This may be because that feature fixation is performed on the token itself where the acquired information is obviously less than that from temporally nearby tokens.

Table 3: Effect of neighbourhood association and feature fixation. "SS", "TS", and "FF" represent spatial shift, temporal shift, and feature fixation respectively. The baseline here is the model with pure linear attention.

| SS | TS | FF | Top-1 |
|---|---|---|---|
| ✗ | ✗ | ✗ | 61.67 |
| ✓ | ✗ | ✗ | 61.99 |
| ✗ | ✓ | ✗ | 64.18 |
| ✓ | ✓ | ✗ | 64.28 |
| ✗ | ✗ | ✓ | 62.57 |
| ✓ | ✓ | ✓ | 65.46 |

Table 4: Ablation on fixation options to different targets. "Sep." means the feature fixation ratios are separately generated. "Coo." is the abbreviation of cooperation. "FS" refers to sharing the fixation ratio. The baseline here is the model that only uses the neighbourhood association.

| Targets | Sep. | Coo. | FS | Top-1 |
|---|---|---|---|---|
| No fixation | ✗ | ✗ | ✗ | 64.28 |
| Q, K | ✓ | ✗ | ✗ | 64.46 |
| Q, K | ✗ | ✓ | ✗ | 64.61 |
| Q, K | ✗ | ✓ | ✓ | 64.78 |
| Q, K, V | ✗ | ✓ | ✗ | 65.33 |
| Q, K, V | ✗ | ✓ | ✓ | 65.46 |

Table 5: Effect of kernel functions. The baseline here is our default full model.

| Kernel | Top-1 |
|---|---|
| ELU+1 [35] | 60.42 |
| Sigmoid | 63.26 |
| ReLU (ours) | 65.46 |

Table 6: Effect of attention types. The baseline here is our default full model.

| Attention | Top-1 |
|---|---|
| Joint [6, 4] | 58.17 |
| Windowed [7] | 61.08 |
| Factorized (ours) | 65.46 |

Table 7: Performance of different model variants.

| Variants | Top-1 | Top-5 | GFLOPs |
|---|---|---|---|
| S | 63.48 | 88.68 | 129.1 |
| H | 65.67 | 90.27 | 530.2 |
| HR | 65.85 | 89.97 | 573.3 |
| Default | 65.46 | 90.11 | 257.8 |

**Fixation options**. We ablate the design options inherent in feature fixation in more details below, and the baseline switches to the model that only uses the neighbourhood association. Since the attention is calculated from Q and K, we start with the setting that Q and K learn to reweight features by themselves without any cooperation, *i.e.*, Eq. 4. From Table 4, we find that the operation only contributes a minor performance gain, *i.e.*, 0.18%. After enabling the cooperation between Q and K, the accuracy is further improved by 0.15% (0.33% better than the baseline). Next, we add V to the cooperation space, *i.e.*, Eq. 5, and the performance is significantly enhanced by 1.05% compared to the baseline. It indicates that simultaneously considering Query-Key-Value is beneficial for discriminating feature importance more comprehensively and appropriately. Sharing the fixation ratio can trigger common feature representations like the weight sharing technique [54, 79], and is also proven to be useful in improving the Top-1 accuracy in both Query-Key and Query-Key-Value cooperation cases. Moreover, it saves ∼0.1M parameters owing to the shared weights in learning the fixation ratio.

In addition to the channel-wise concatenation in our implementation, we also compare other typical cooperation manners for Query-Key-Value feature aggregation, *i.e.*, element-wise addition and multiplication. Both of them lead to ∼0.5% drop in Top-1 accuracy, so feature concatenation is our best option in this case with only a minor increase in the parameter scale (∼0.05M).

**Kernel function.** We study the options of kernel functions in Table 5. Without a kernel function, the network cannot converge properly. As indicated by [60, 35], the functions should be non-negative to facilitate the aggregation of more positively-correlated features. We try two other typical non-negative functions, *i.e.*, ELU+1 [35] and Sigmoid, but their performance is clearly inferior to ReLU.

**Attention pattern.** By default, we employ the factorized spatial-temporal attention introduced in [6, 4]. Specifically, the attention is first computed spatially (tokens from the same frame) and then temporally (tokens from different frames at the same spatial location). For comparison, we also test another two popular attention patterns, *i.e.*, joint attention [6, 4] and windowed attention [7]. As shown in Table 5, our choice of using factorized attention yields the best performance. Joint attention is less comparable as it takes all tokens into account, inevitably containing more redundant and noisy features. Windowed attention only aggregates local temporal information, which is less sufficient than ours that collects tokens from the entire temporal domain.

**Model variants.** In addition to our default model, we provide three extra variants: 1) *S*: a shorter video clip as $8 \times 224 \times 224$; 2) *H*: a longer video clip as $32 \times 224 \times 224$; and 3) *HR*: a larger spatial resolution as $16 \times 336 \times 336$. The results are displayed in Table 7. With longer video input or a larger spatial resolution, the classification accuracy can be further improved because more

Table 8: Throughput evaluation measured by *videos/second*. A larger value means higher speed. Our model maintains good efficiency in various settings of input frames and spatial resolution.

| # Frames | 16 | | | 32 | | | 64 | | | 96 | | |
|---|---|---|---|---|---|---|---|---|---|---|---|---|
| Resolution | **224** | **336** | **448** | **224** | **336** | **448** | **224** | **336** | **448** | **224** | **336** | **448** |
| Orthoformer [57] | 1.79 | 1.59 | 1.41 | 1.71 | 1.56 | 1.15 | 1.53 | 0.97 | OOM | 1.25 | 0.83 | OOM |
| Ours | 10.25 | 8.57 | 5.40 | 8.38 | 4.68 | 3.22 | 6.58 | 3.75 | 2.81 | 4.48 | 2.30 | OOM |
| Softmax | 8.23 | 6.49 | 4.57 | 6.15 | 4.57 | 2.57 | 5.27 | 2.55 | OOM | 4.27 | OOM | OOM |

temporal or spatial information is included. However, this always comes with a clear increase in computational costs, *e.g.*, GFLOPs (also see Table 8 for throughput comparison). Our default model can better balance the accuracy and efficiency. In real-world applications, end-users can use any variant as per their practical needs.

**Efficiency analysis.** To measure the actual computational efficiency, we report the throughput (*videos/second*) in Table 8. For a fair comparison, the Orthoformer attention [57] (linear) and Softmax are incorporated into our backbone as a replacement of the proposed attention. All models are tested on a 80G A100 GPU card with a batch size of 1, and their throughput is the average of ten runs. Our model achieves the best efficiency in various settings of input video frames and their spatial resolution. Remarkably, our good efficiency is maintained even with a larger video input, *e.g.*, $64 \times 448 \times 448$ and $96 \times 336 \times 336$, where the Orthoformer and Softmax tend to be out of memory (OOM). Note that the Orthoformer attention presents significantly lower efficiency, which is also observed in [93] and may be caused by the serial computation in landmark selection. It is natural that all models are OOM when the input scale is extremely large, *e.g.*, $96 \times 448 \times 448$. This can be partially solved by token sparsification [63, 27], and we leave it to the future work.

**Application to existing models.** Lastly, we study the effect of applying neighbourhood association and feature fixation to other linear attention and Softmax in Table 9. The two modules can clearly improve the performance of linear attention, *e.g.*, Performer [14], Linear Trans. [35] and cosFormer [60], and their combination makes the three linear Transformers even closer to the SOTA results. It validates the effectiveness of our method in the linear setting. Nevertheless, there is no performance gain in the Softmax-based Transformer,

Table 9: Application to existing models. We report the Top-1 accuracy. "NA" and "FF" abbreviates for neighbourhood association and feature fixation respectively.

| Method | Baseline | +NA | +FF | +NA+FF |
|---|---|---|---|---|
| Performer [14] | 58.82 | 61.57 | 59.97 | 63.50 |
| Linear Trans. [35] | 60.42 | 62.84 | 61.87 | 64.76 |
| cosFormer [60] | 61.18 | 63.23 | 61.94 | 64.97 |
| XViT (Softmax) [60] | 66.20 | - | 65.94 | 65.94 |

*e.g.*, XViT [7] (neighbourhood association is not added here because XViT already has the temporal feature shift operation). This is not surprising because it further reveals that our method is specially useful to linear attention.

## 6 CONCLUSION

In this paper, we study the linear video Transformer for video classification to reduce the quadratic computational costs of standard Softmax attention. We analyze the performance drop in linear attention, and find that the lack of attention concentration to salient features is the underlying cause. To deal with this issue, we propose to use feature fixation to the query and key prior to calculating the linear attention. To measure feature importance more comprehensively, we facilitate the cooperation among Query-Key-Value by aggregating their features when generating the fixation ratio. Moreover, motivated by the fact that salient vision features are usually locally accumulative, we apply the feature shift technique to get additional feature guidance from spatially and temporally nearby tokens. This, combined with feature fixation, produces state-of-the-art results among linear video Transformers on three popular video classification benchmarks. Our performance is also comparable to some quadratic Transformers with Softmax attention with fewer parameters and higher efficiency. Our future work will focus on further accelerating the linear attention model without degrading the general performance, *e.g.*, token sparsification, network pruning, and so on.

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

# A  APPENDIX

## A.1  TRAINING DETAILS

We provide detailed training configuration on SSv2 and K400/K600 in Table 10.

Table 10: Detailed training configuration of SSv2 and K400/K600.

|  | SSv2 | K400/K600 |
|---|---|---|
| Batch size | 32 | 32 |
| Video sampling rate | 16 | 16 |
| Training jitter scales | $256 \times 320$ | $256 \times 320$ |
| Hidden dimension | 512 | 512 |
| Base learning rate | 0.035 | 0.035 |
| LR policy | cosine | cosine |
| Max epochs | 35 | 35 |
| Weight decay | 0.0004 | 0.0004 |
| Warm-up epochs | 5 | 5 |
| Warm-up start LR | 0.003 | 0.01 |
| Optimizer | sgd | sgd |
| Dropout rate | 0.5 | 0.5 |
| Spatial shift | 1 | 4 |
| Temporal shift | 4 | 8 |

## A.2  FURTHER ILLUSTRATION OF FEATURE FIXATION

To illustrate the proposed feature fixation, we give a toy example here. Suppose we have two values, $a$ and $b$, from the multiplication of a specific query to two keys, *i.e.*, $a, b > 0$, and $b$ is assumed to be more important. In linear attention, the original score of $b$ is $\frac{b}{a+b}$, *i.e.*, dot-product normalization. Now we reweight $a$ and $b$ into $m_1 a$ and $m_2 b$, where $m_1, m_2 \in (0, 1)$ and $m_1 < m_2$. Consequently, the updated score of $b$ is $\frac{m_2 b}{m_1 a + m_2 b}$. In that case, $\frac{m_2 b}{m_1 a + m_2 b} / \frac{b}{a+b} = \frac{a+b}{\frac{m_1}{m_2} a + b} > 1$, *i.e.*, $\frac{m_2 b}{m_1 a + m_2 b} > \frac{b}{a+b}$, indicating that the impact of $b$ is enhanced.

In our feature fixation, each feature channel is assigned with different weighting factors (ranging from 0-1), where non-essential features are down-weighted and critical features are almost unchanged. Hence, some multiplication values of the reweighted query and key become smaller if they contain non-essential features. In that case, their final scores are also reduced so that the impact of other important scores is further enhanced in turn (the principle of the toy example above is also applicable here).

## A.3  FEATURE SHIFT

**Temporal feature shift.** We define a temporal window $[-\tau, \tau]$ centred at the target token, similar to [7]. When reconstructing the key/value vector of a patch located at a specific position in that frame, we first retain its first $\alpha D$ feature channels where $\alpha \in [0, 1]$ is a scaling factor and empirically set as 0.5. The remaining part is filled by the feature concatenation of its nearby $2\tau$ key/value vectors, each sequentially contributing $(1 - \alpha)D/2\tau$ channels.

**Spatial feature shift.** Since the video frame is a 2D image, an intuitive way is to define a squared kernel centered at the target token and set the spatial shift size (*i.e.*, the radius of the kernel) as $\xi$. In that case, $(2\xi + 1)^2$ tokens are involved to construct the new key/value vector. However, when the shift size is large, *i.e.*, enlarging the receptive field, the quadratic increase of the token number would weaken the impact of each nearby token as only very limited features are transmitted. Inspired by the criss-cross attention [33] and one-way spatial affinity [48, 47] in CNNs, we only consider $4\xi$ tokens in the same column and row of the target, *i.e.*, $\xi$ tokens from each direction (left, right, above, below). Similar to the temporal shift, each nearby token supplies $(1 - \alpha)D/4\xi$ feature channels.

To verify this, we compare the squared kernel shift and spatial shift with different sizes in Table 11. As shown, when the size is 1, the numbers of tokens are similar and their performance is identical.

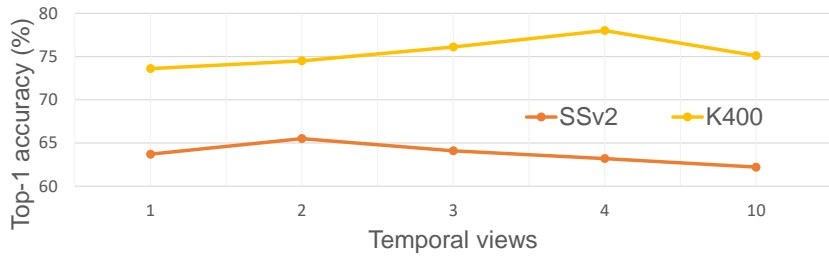

Figure 4: Impact of temporal views at the inference stage. The performance varies with the number of temporal views on both SSv2 and K400. We can empirically choose the best setting from various choices according to the performance on each dataset.

However, with the increased size, the token number grows quadratically in the squared kernel but the performance is degraded. Therefore, the spatial shift we used can fulfill our needs, *i.e.*, allowing the target token to communicate with sufficient nearby tokens without significant performance degradation caused by involving excessive nearby tokens.

Table 11: Comparison of spatial shift (ours) and squared kernel shift on SSv2. The baseline here is our full model.

|  | Size | #Tokens | Top-1 | Top-5 |
|---|---|---|---|---|
| Spatial shift (ours) | 1 | 5 | 65.5 | 90.1 |
| Squared kernel shift | 1 | 9 | 65.5 | 90.1 |
| Spatial shift (ours) | 2 | 9 | 65.4 | 90.0 |
| Squared kernel shift | 2 | 25 | 65.2 | 89.8 |

## A.4 IMPACT OF TEST VIEWS

At the inference stage, the number of spatial views are normally set as 3, *i.e.*, left-crop, middle-crop, and right-crop. As shown in Table 1 and 2, almost all the models employ this setting of spatial views, so we do not ablate it here. We study the effect of temporal views in Fig. 4. The performance varies with the number of temporal views on both SSv2 and K400. We can empirically choose the best setting from various choices according to the performance on each dataset.

## A.5 ABLATION ON K400

Lastly, we supplement the ablation study on the effect of neighbourhood association and feature fixation on K400 in Table 12. Compared with the results of SSv2 (Table 3), the Top-1 accuracy does not vary too much on K400, but the effectiveness of each module is well validated. A distinction here is that the spatial shift brings more performance gain than the temporal shift. This is because different from SSv2 that reasons more about motion cues, K400 focuses more on spatial content as it has more complicated background information [4, 6, 50].

Table 12: Effect of neighbourhood association and feature fixation on K400. "SS", "TS", and "FF" represent spatial shift, temporal shift, and feature fixation respectively. The baseline here is the model with pure linear attention.

| SS | TS | FF | Top-1 |
|---|---|---|---|
| ✗ | ✗ | ✗ | 77.27 |
| ✓ | ✗ | ✗ | 77.53 |
| ✗ | ✓ | ✗ | 77.43 |
| ✓ | ✓ | ✗ | 77.70 |
| ✗ | ✗ | ✓ | 77.47 |
| ✓ | ✓ | ✓ | 77.96 |

