# OpenReview forum: "Linear Video Transformer with Feature Fixation"
_ICLR.cc/2023/Conference — Submitted to ICLR 2023_

### Official Review · Reviewer_uvWv · 2022-10-20

**Confidence:** 4
**Clarity, Quality, Novelty And Reproducibility:** 1 Clarity

The paper is well-written.…
**Correctness:** 3
**Technical Novelty And Significance:** 2
**Empirical Novelty And Significance:** 2
**Recommendation:** 3

**Strength And Weaknesses:**

1 Strength

1) The computation problem of ViT is a key issue for video modeling.

2) The paper is well-written.

2 Weakness

1) Novelty. The novelty is limited.

First, adding such fixation (or gating) mechanism has been widely used in CNN, e.g., SE or CBAM. I understand the detailed difference between this paper and other gating method. But the key idea or design is similar.

Second, temporal shift operation has been proposed in TSM.

2) Results. The results are not quite convincing.

First, for SSV2, there are many better solutions which are ignored in this paper, e.g., (MViT-B, Top1:67.1, Top5:90.8, GFLOPs: 170×3×1), (UniFormer-B, Top1:70.4, Top5: 92, GFLOPs: 290×3×1), etc. Linear attention based ViTs are not the popular or well-known approaches in video modeling. I suggest to compare with the well known efficient models in video recognition literature. For Kinetics, please report the GFLOPs to evaluate computation cost.

Second, it is not convincing if the feature fixation is the most important design in this work. From table 3, one can easily see that, temporal shift is the key operation. However, this shift operation has been proposed in TSM. Hence, the contribution in this work may not be convincing.






**Summary Of The Paper:**

In this work, the authors propose to alleviate computation cost of ViT for video modeling. To achieve this goal, they use linear attention with a feature fixation module. Specifically, they introduce two types of fixation by gating or attention on feature channels. Moreover, they use spatial and temporal shift for feature association.

**Summary Of The Review:**

Overall, this paper proposes to reduce computation burden for video modeling. The problem is important. But the fixation design is limited. The experiments are not quite convincing too.

---

### Official Review · Reviewer_xSk5 · 2022-10-24

**Confidence:** 4
**Correctness:** 4
**Technical Novelty And Significance:** 2
**Empirical Novelty And Significance:** 2
**Recommendation:** 3

**Clarity, Quality, Novelty And Reproducibility:**

Clarity could be improved. In particular, I find the use of some terms, such as "cooperative feature fixation" for feature reweighting or "neighborhood association" for shifting, confusing and making reading unnecessarily harder.

Quality of the experiments can be improved. I think a more detailed discussion on pretraining and adding missing prior works for Table 1 and showing flops & params in Table 2 would help.

I think the novelty is ok.

**Strength And Weaknesses:**

Strengths:
+ Reducing the computational cost for video models can potentially be highly impactful due to the very high computational cost of most video models.
+ Bringing expressiveness to linear transformer to make it work better makes sense.

Weaknesses:
- The improvement brought by feature fixation is small (1.2%). I'm not sure if the gain worths the complexity and additional computations added to the model.
- Table 3 shows that the component that brings most of the improvement is "temporal shift", which is proposed in prior work (e.g. [45, 51]), instead of the proposed feature fixation. Thus, even if the overall method outperforms some prior works, it's not convincing that it's due to the proposed method.
- Models compared in Table 1 use different pre-training datasets. In particular, the proposed method uses a strong IN21K pre-training, which could put prior works that do not at a disadvantage. Thus overall, without discussing on the effect of pre-training datasets, it's hard to draw conclusions from the table.
- Some state-of-the-art results are missing. For example, Swin-B achieves 69.6 and MViTv2 achieves 73.3 accuracy on SSv2, which are both significantly higher than the 65.5 presented, but are not compared with.
- minor points on writing: I'm not a big fan of using a new term to describe something that's already used. For example, prior works have widely used "excitation" ([31]) and "gating" ([55]) for the "fixation" operators. I find using new terms make reading more difficult.


**Summary Of The Paper:**

In this paper, the authors study how to close the performance gap between a linear transformer and a standard (softmax) transformer. The motivation is that linear transformer can be computationally beneficial when the sequence length is high. The main proposed method is called "feature fixation", which re-weights the Query and Key in channel dimension in attention. The authors perform a few ablation studies and compare the proposed method with a number of prior works.


**Summary Of The Review:**

Overall, my major concern is that the experiments do not support the usefulness of the method. The small improvement brought by feature fixation is unfortunately a bit disappointing.

---

### Official Review · Reviewer_8R83 · 2022-10-24

**Confidence:** 4
**Correctness:** 4
**Technical Novelty And Significance:** 2
**Empirical Novelty And Significance:** 3
**Recommendation:** 6

**Clarity, Quality, Novelty And Reproducibility:**

The paper is clear and should be reproducible from the provided details. The approach is not especially novel, being a combination of existing linear attention and re-weighting approaches. However, this combination is effective, which is valuable.

**Strength And Weaknesses:**

The paper is well written, and the experiments are well done, showing the benefit of the approach. The approach is largely a combination of existing components, however, this combination works. The experiments and ablations are thorough and show the benefit of all the piece of the approach.

However, some of the results are concerning. For example, T3 seems to suggest that the shift is beneficial, rather than the reweighting. There's also some differences in the setup of model training in T1 (e.g. pretraining datasets, learning rates settings, etc), which makes it harder to tell if the benefit of this approach is the pretraining or the model/method.


**Summary Of The Paper:**

This paper proposes a method for more efficient video transformer networks based on linear attention. The main contribution is using a context-gating like mechanism to re-weight the features before attention. The approach is evaluated on kinetics and something-something datasets.

**Summary Of The Review:**

The experiments show the benefit of the approach, and it reduces the FLOPs vs. softmax attention, while still having strong performance. The approach is not greatly novel, but the combination of components works, which is a valuable contribution.

---

### Official Review · Reviewer_nema · 2022-10-28

**Confidence:** 1
**Clarity, Quality, Novelty And Reproducibility:** See above.
**Correctness:** 3
**Technical Novelty And Significance:** 2
**Empirical Novelty And Significance:** 3
**Recommendation:** 5

**Strength And Weaknesses:**

The following are some detailed questions and comments about the paper:

1, Does the paper introduce additional parameters to the model, such as in Eq. (4)? The description in section 4.1 can be more clear, especially which parameters are learnable.

2, The throughput improvement over softmax seems to be limited based on results from Table 8. However, the theoretic FLOP saving is more significant from Table 1. Is this due to not very well optimized implementation of the proposed method? Please provide more insight if possible.

3, The proposed method is very generic. But the paper doesn’t provide results to prove how well the proposed method can generalize to other cases. For example, can we apply the method to other backbones, such as TimeSformer, MViT, etc? Can we apply the method for image classification as well? If the proposed method is coupled with a specific backbone, such as XViT, the impact of the paper will be less significant.

4, In Table1, why use 2x3 views for the proposed method, and 1x3 views for XViT? Since the proposed method is using XViT as the backbone, it should use the same setting. Will the performance gap between the proposed method and XViT become much larger, if they both use 1x3 views?


**Summary Of The Paper:**

The paper focuses on improving the efficiency of standard vision transformers, which have quadratic computational complexity. The paper follows previous methods using linear attention, which does reduce the computational complexity, but results in worse performance. The paper proposes to mitigate the performance drop caused by linear attention with feature fixation, and also improves feature fixation by neighborhood association. The paper demonstrated the effectiveness of the proposed method on SSv1 and SSv2, Kinetics400 and Kinetics600.


**Summary Of The Review:**

See above.

---

### Decision · Program_Chairs · 2023-01-20

**Decision:**

Reject

**Justification For Why Not Higher Score:**

- Main claims are not validated (is the shift bringing the performance benefit or the proposed feature fixation module)?
- Modest empirical improvements, issues with empirical setup raised by the reviewers
- Limited novelty

**Justification For Why Not Lower Score:**

N/A

**Metareview: Summary, Strengths And Weaknesses:**

The authors propose to alleviate the quadratic complexity in Vision Transformers, specifically in the context of action recognition in video data where this issue is critical. Standard approaches which replace the classic attention layer with linear attention offer a speedup, but may result in a large gap in accuracy. The authors continue this line of work and aim to close this gap by introducing a feature fixation module which re-weights the feature importance of the query and key prior to computing linear attention. The empirical evaluation is performed on Kinetics and Something-something where the method clearly reduces FLOPS while achieving competitive performance.

The reviewers appreciated the fact that reducing the computational complexity of transformers in the context of video understanding is a key challenge. However, the reviewers pointed out that the obtained improvements are too modest and that it's unclear whether the improved performance stems from the proposed mechanism or other cofounding factors. Given the modest improvements and lack of novelty, I will recommend the rejection of this manuscript. Nevertheless, I hope the authors continue this line of work and provide more compelling evidence for the proposed approach.